# Hyperplanar Morphological Clustering of a Hippocampus by Using Volumetric Computerized Tomography in Early Alzheimer’s Disease

**DOI:** 10.3390/brainsci7110155

**Published:** 2017-11-21

**Authors:** Sarawut Suksuphew, Paramate Horkaew

**Affiliations:** 1School of Medicine, Institute of Medicine, Suranaree University of Technology, Nakhon Ratchasima 30000, Thailand; 2School of Computer Engineering, Institute of Engineering, Suranaree University of Technology, Nakhon Ratchasima 30000, Thailand; phorkaew@sut.ac.th

**Keywords:** early Alzheimer’s disease, volumetric computerized tomography, shape analysis, support vector machine

## Abstract

**Background**: On diagnosing Alzheimer’s disease (AD), most existing imaging-based schemes have relied on analyzing the hippocampus and its peripheral structures. Recent studies have confirmed that volumetric variations are one of the primary indicators in differentiating symptomatic AD from healthy aging. In this study, we focused on deriving discriminative shape-based parameters that could effectively identify early AD from volumetric computerized tomography (VCT) delineation, which was previously almost intangible. **Methods**: Participants were 63 volunteers of Thai nationality, whose ages were between 40 and 90 years old. Thirty subjects (age 68.51 ± 5.5) were diagnosed with early AD, by using Diagnostic and Statistical Manual of Mental Disorders IV (DSM-IV) criteria and the National Institute of Neurological and Communicative Disorders and the Stroke and the Alzheimer’s disease and Related Disorders Association (NINCDS-ADRDA) criteria, while the remaining 33 were in the healthy control group (age 67.93 ± 5.5). The structural imaging study was conducted by using VCT. Three uninformed readers were asked to draw left and right hippocampal outlines on a coronal section. The resultant shapes were aligned and then analyzed with statistical shape analysis to obtain the first few dominant variational parameters, residing in hyperplanes. A supervised machine learning, i.e., support vector machine (SVM) was then employed to elucidate the proposed scheme. **Results**: Provided trivial delineations, relatively as low as 5 to 7 implicit model parameters could be extracted and used as discriminants. Clinical verification showed that the model could differentiate early AD from aging, with high sensitivity, specificity, accuracy and F-measure of 0.970, 0.968, 0.983 and 0.983, respectively, with no apparent effect of left-right asymmetry. Thanks to a less laborious task required, yet high discriminating capability, the proposed scheme is expected to be applicable in a typical clinical setting, equipped with only a moderate-specs VCT.

## 1. Introduction

Alzheimer’s disease (AD) comprises of several risk factors [1,2,3], including not only aging process but also a genetic predisposition that causes changes in specific nerve cells and accumulation of neuritic plaques and neurofibrillary tangles [4,5]. It is often necessary for a physician to conclusively identify the presence of AD, especially in its early stage. The clinical diagnosis of AD must therefore be considered based upon both history and symptom progression to be differentiable from other secondary dementias [6,7,8,9,10].

Structural neuroimaging in dementia such as Computed Tomography (CT) or Magnetic Resonance Imaging (MRI) of the brain is generally adopted in differential diagnosis by determining the anatomical plane and location of atrophy. As for AD diagnosis, recent studies [11,12,13] have shown that structural imaging could play a key part in evaluating medial temporal atrophy, width of choroidal fissure, width of temporal horn and height of hippocampal formation. Typically, MRI is preferred to CT, because it has the advantage of being able not only to assess regional atrophy but also to show detailed changes such as white matter lesions and microbleeds. Unfortunately, state of the art MRI is not available in most hospitals in developing countries. Alternatively, there are others using volumetric CT (VCT) in assessments of cerebral atrophy, regional volumetric measurements, and parenchymal density measurements [14,15,16]. Although there exist hippocampal ratings based on Scheltens’ scale, this visual analog assessment is rather subjective and can be made accurately only by an expert neurologist. Moreover, differentiating a normal from early stage AD by Schelten’s scale reading calls for prominent and thoroughly definitive boundaries available only in MRI. Preliminary screening by using a more cost effective CT, before referring to MRI study, is therefore a viable yet challenging alternative.

Thus far, the structural-based schemes share similar hindrances. Firstly, definite line has to be drawn empirically on a set of selected measurements to separate symptomatic attributes from the asymptomatic ones. This inevitably causes large inter-reader variabilities and dependencies on various other factors, e.g., ethics and other clinical exposures. Moreover, predefined structural criteria become less definitive as the disease is yet in early development, making assessments based on imaging interpretation even more dubitable.

Existing morphological analyses typically rely on imposing some statistical distributions over Euclidean-based measurements, e.g., length, area, or volume, etc. This sample space was then divided into that of normal and symptomatic, along a discrimination line between respective modal curves. Unlike those schemes, in which metric assumptions were made a priori, Statistical Shape Analysis (SSA) computationally deduces mutual variations of geometrical properties within the sample space. The most discriminative separations are then implicitly made along axes containing the largest variations. More specifically, SSA first calculates a co-variances matrix of all corresponding control points describing the shape. This matrix is then decomposed by using Eigen bases into a set of spanning vectors, called principal components (PC), arranged by their importance [17,18]. With the SSA, a Mahalanobis distance between respective PCs is used to measure the difference between any shape pair. Accordingly, their cluster separations can be intuitively determined, without having to compare some measurements arbitrarily. Due to this preferable characteristic, the SSA has been widely employed in various studies, including medical imaging [19,20].

Image analysis of the hippocampus has currently been a topic of interest in diagnosis of AD [21,22,23]. Provided a set of hippocampal shapes, those techniques created a generalized model of a hippocampus, which can be used to distinguish normal from pathological AD hippocampi. More specifically, Mrzilkova et al. [21] manually analyzed MR images and confirmed that hippocampal volume decrease is a primary determinant of AD diagnosis, but found no clinical use of pons and cerebellum volume nor left-right asymmetry. Wang et al. [22], computed hippocampal surface deformation also from MRI but applied SVD to determine different patterns between normal and AD groups. Likewise, Scher et al. [23] compared volumetric MRI of the normal and AD groups, but based on normalized medial axes locally describing the shape. These techniques had reached the similar conclusion that MRI could be employed to derive structural attributes, differentiating AD subjects from healthy aging. They, however, required presumptions regarding the attributes (e.g., volume and local deformation) on which the discrimination is made. Some of these techniques also required tedious manual delineation of entire 3D hippocampi so that sufficiently dense data could be sampled. Furthermore, none of those findings were able to confirm mild symptomatic hippocampi during their early AD stage.

To address those limitations, the aim of this study is to employ VCT for assessing hippocampal structures and then to determine compact yet definitive early AD discriminants, with high sensitivity, accuracy, and precision. Given a set of manually drawn contours on a single coronal slide, we shall later show in the experiments that a relatively small number of parameters could be derived and effectively employed for this purpose. Since the morphological analysis was performed on a single slide, without requiring any tissue intensities, the VCT acquisition was thus preferred to MRI as it induced less tiredness and anxiety for aging subjects. Moreover, apart from hippocampal outline and some fiducial markers, subsequent computations were automatic and required no additional empirical knowledge from neurologist readers. The remainder of the paper is organized as follows. Section 2 explains related materials and the proposed method, including data, analytical procedure and its statistical assessments. Section 3 and Section 4 report experimental results in vivo, compared to conventional metric analysis and their discussions, respectively. Section 5 states the concluding remarks of the current study.

## 2. Methods

### 2.1. Subjects 

From January to March 2015, the participants were recruited from an outpatient clinic at our University Hospital. Cranial VCT scans were acquired from 30 subjects with early AD and from 33 healthy, age-matched subjects in control group. The patients were diagnosed with AD by using DSM-IV criteria and the National Institute of Neurological and Communicative Disorders and the Stroke and the Alzheimer’s disease and Related Disorders Association (NINCDS-ADRDA) criteria in probable level [24,25]. Questionnaires were used to collect the information of age, sex, highest of educational degree, occupation and family history of dementia. Clinical data such as blood pressure, neurological examination record and Mini-mental State Examination in Thai language (TMSE) score were obtained from all subjects (taken into account their differing educational backgrounds). Those subjects with histories of head injury, brain hypoxia, alcoholism, underlying diseases or other involving factors that may affect the size of hippocampus, for instance epilepsy or limbic system disorders were excluded. Written informed consent had been taken from all subjects prior to the study. The study protocol was approved by the ethics committee of Suranaree University of Technology (EC-57-23), conforming to the ethical principles of the Declaration of Helsinki.

### 2.2. Imaging Protocol

The imaging examination was performed by using a two–row spiral CT unit (Dual HiSpeed, GE Medical Systems, Chicago, IL, USA). An unenhanced CT images of the head was first acquired in sequence with slice thicknesses of 7 mm in supratentorial space and of 4 mm within the posterior cranial fossa. Contrast enhancing agent was later administered and then another set of CT scans was acquired with a slice thickness in axial direction of 0.625 mm and reconstructed with 2.5 mm resolutions in axial, coronal and sagittal views. The regions of interest (ROI) were fixated around hippocampal outlines (Figure 1). 

### 2.3. SSA Software

The software used in the subsequent analyses was developed in-house specifically for the purpose of this study. It supports Picture Archiving and Communication System (PACS) standard by using the DCMTK™ (NEMA 2016, OFFIS 2011) library and written in C++ language. This software (SSA 1.0, Suranaree University of Technology, NR, Thailand) consisted of two modules, i.e., the expert training and the automatic morphological analysis. The purpose of the first module was to aid the reader to browse and load a set of enhanced CT picture from a selected study stored in the Digital Imaging and Communication in Medicine (DICOM) format on a PC panel. This module also enabled interactive manipulation of imaging data, including slice selection, window/level and scale adjustments, and panning so that a hippocampal area of interest best fitted to the viewing frame. The radiological reader then utilized the built-in drawing tools to delineate both sides of hippocampi, manually based on prescribed topology (Figure 2). 

The second module analyzed the hippocampal shapes from the training set and constructed a statistically deformable model describing their plausible anatomical variations. Specifically, since these shapes were drawn free hand and as such were difficult to correlate their control points, they were firstly fitted with a cubic spline and regularized with respect to the prescribed fiducial markers, by using spline arc-length re-parameterization [26]. Subsequently, each regularized hippocampus was expressed as a vector of concatenated coordinates (*x*, *y*) of *N* control points, as given in Equation (1), where *i* is the shape index.

(1)$$x_{i}={\left[x_{i1},y_{i1},x_{i2},y_{i2},\ldots ,x_{ik},y_{ik},\ldots ,x_{iN},y_{iN},\right]}^{T},$$

Once their correspondence was established, the shapes were aligned by Procrustes analysis [27] to remove geometrical biases due to size, locations and orientation. These normalized co-registered dataset were then analyzed with Principal Component Analysis (PCA) to produce a Karhunen-Loeve expansion of bases describing plausible dominant variations found in the training set, with respect to their mean shape and control points covariance, as given by Equation (2), where **x** is a shape vector whose index was *i* = 1 … *N*, $\bar{x}$ is the mean shape, **P***_S_* and **b***_S_* are the *2N* Eigen bases *2N* dimensions and shape parameters vector of *2N* elements, respectively.

(2)$$x_{i}=\bar{x}+P_{s}b_{s},$$

The model expressing a hippocampal instance by these linear combinations of the bases, **P***_S_*, is hereon referred to as Statistical Shape (SS). It would be shown later in the experiments that due to favorable properties of the PCA, any incremental contribution due to an additional basis decreased as a more number of modes were included. In other word, majority (2–3 times the range of multi-dimensional standard deviations) of statistical distribution could be sufficiently captured by the first few modes of variations. Accordingly, a hippocampal instance may be synthesized faithfully (i.e., conforming to the hippocampal anatomy) by as few (*M*) truncated model parameters, as given by Equation (3), where $P_{s}^{m}$ is the *m*th bases vector in the shape bases and $b_{s}^{m}$ is the corresponding shape parameter, respectively. The number of modes *M* was computed such that accumulated variance up to the value *M* comprised required variation (e.g., within ±3σ) found in the training set.

(3)$$x_{i}=\bar{x}+\sum_{m=1}^{M}P_{s}^{m}b_{s}^{m},$$

### 2.4. Supervised Model Clustering

Truncated SS model parameters extracted per each pair of hippocampi with associated labels of both AD and non-AD groups were fed into a Support Vector Machine [28] (SVM) for supervised classification. A given sample *x* was classified based on their corresponding PCA model parameters *b* by using the discrimination functions, given in Equation (4), where *sgn* is the sign function, *K* is a linear kernel function of *s*, whose slope and interception point were computed during the training process and the magnitude $\alpha_{i}$ and bias values *b* were empirical SVM parameters [29], set by default.
(4)$$f\left(x\right)=sgn\left[\underset{i}{\sum }\alpha_{i}y_{i}K\left(x_{i}\cdot x_{j}\right)+b\right]$$
K(s)=γs+β

### 2.5. Benchmarking with Conventional Metric Analysis

It is clinically established that one of the major determinants for diagnosing AD in general is hippocampal formation area. By visual inspection, an expert reader needs to consciously correlate this area with other peripherals. For screening purposes, however, some readers prefer simplifying this measurement to formation height (Figure 2). These procedures have raised some concerns, namely, subjective bias due to overall head size and variability of reference axes. To elucidate this statement and thus highlight the merits of this study, the proposed morphological clustering was benchmarked against that based on relative formation area, whose value was computed using Trapezoidal integral.

### 2.6. Validation and Performance Analysis

Fisher exact tests and Student *t-*tests were used to compare between demographics and clinical characteristics. The hippocampal shape classification was evaluated by using statistical measures [30]. Specifically, the precision (producer’s accuracy) is defined as the ratio between the samples correctly predicted as having AD (true positive, TP) and those predicted as such (true and false positives, TP+FP). The recall (user’s accuracy or sensitivity) is defined as the ratio between the samples correctly predicted as having AD and those in fact were diagnosed with AD (true positive and false negative, TP+FN). The accuracy is the proportion of the correct results among the total number of instances examined and was defined as the ratio between the samples correctly predicted as and as not having AD (true positive and negative, TP+TN). F-measure is defined as the harmonic mean of precision and recall.

## 3. Results

Sixty-three participants were divided into two groups, whose details of baseline characteristics were given in Table 1.

### 3.1. Hippocampal Morphology Variability

The readings of hippocampal morphology from two out of three readers were shown (Figure 3). It is clear that there existed significant intra- and inter-observer variabilities. Particularly, a mild AD case would have been diagnosed differently by reader A and B if they had done based primarily on, for instance, hippocampal formation area (top row). On the other hand, depending on an empirically defined threshold, readers A and B might as well or might not agree on the other case (bottom row) with less noticeable discrepancies. These ambiguities thus often call for neurological experts to confirm the result.

The initial Degree of Freedom (DoF) of a hippocampal contour per each side was 40, i.e., 2 for (*x*, *y*) coordinates multiplied by 20 control points. A total of 63 samples were assessed independently by three readers, each repeated their tracing 2–3 times, hence a total of 382 delineated instances were produced. These instances were analyzed with PCA and their principal modes of variations were extracted. Out of the 40 DoFs, only 9 modes (i.e., model parameters) were able to capture 95% of total variations. The variational patterns found in the first two significant model parameters are depicted in Figure 4, showing the respective variations within ±2 standard deviations.

To determine whether the Statistical Shape (SS) model parameters were capable of separating control from AD subject groups, their scattered plots were evaluated. For clarity, only the first two dominant modes of all 382 instances were shown (Figure 5). Each point corresponds to a shape instance drawn by one reader on one subject. A reader might repeat his/her delineations for any subject a couple times, depending on his/her confidence. Both groups were clustered with some noticeable outliers and overlapping. This is due to that only 2 out of 10 modes were considered, leaving out 8 discriminant factors, which would have been comprising 95% of total variations.

The former issue could trivially be elevated by including even more number of modes into classification, whose results will be reported in the next section. The ambiguity due to inter and intra observer variabilities was resolved by averaging readings in the model space. The resultant scattered plot therefore consists of only 63 consensus points, corresponding to 30 and 33 controls and subjects (Figure 6). With inter and intra observer variabilities being reduced, the cluster separation become more apparent, leaving merely a few overlaps near the boundary. 

### 3.2. Multi-Dimensional SVM Classification

To assess the model performance, an SVM with linear kernel were adopted to perform clustering based on first three model parameters, resulting in a 3D hyperplane separating both groups. All 63 samples annotated as controls and subjects were used as the training set. By using the SVM linear classifiers, the respective numerical assessment results are listed in Table 2. It is evident that, with only three model parameters, the misclassification errors were only a few samples per either side, i.e., 95.24 and 98.41 percent, respectively. In addition, there was no need to use higher order kernel (e.g., polynomial or RBF, etc.) as a linear support vector already satisfactorily divided both classes.

### 3.3. Optimal Number of Model Parameters

We have analyzed the training set and found that 95% of total morphological variations can be captured by ten model parameters. In statistical perspectives, this means that based on these data, (1)Both seen and unseen samples can be described by a linear combination of only 10 bases (compactness and generalization abilities of the model).(2)Any linear combination generated by randomized (with a Gaussian distribution) model parameters can synthesize a sample that is closely resemble to those previously seen (specificity) [31].

Care should therefore be observed when applying these model parameters for AD classification. On one hand, as highlighted in the previous sub-section, taking into account only a few parameters could result in indecisive areas. On the other hand, Occam Razor’s principal [32] stated that if there are more than one explanations (in our case, the number of model parameters) for an occurrence (AD condition), the simplest one prevails. Relying on relatively small sample size, an over complicate model could adversely cause over fitting. An experiment was hence carried out to find the optimal model for detecting mild case in early AD. To this end, the first *N* model parameters were fed in to an SVM in turn. In each scenario, the corresponding TP, TN, FP, FN for healthy controls (C) and AD subjects (S) were computed (Table 3). 

## 4. Discussion

Although the VCT images were contrast-enhanced, hippocampal morphology and their boundaries remained subject to visual interpretation by the readers. Bi-laterality and hemispheric asymmetry are widely recognized as the indispensable evolutionally conserved attributes of the brain [33]. In particular, the hemispheric asymmetry of the healthy human hippocampus is well established [34,35]. These factors could especially affect and undermine the assessment of brain structures. In this study, an attempt had thus been made to specifically exclude asymmetric subjects by examining their history, following the selection protocol [24,25]. However, insignificant hippocampal L–R asymmetry was noticeable in Figure 4. Indeed, since both shape and SVM models were built for each side of hippocampi separately, slight differences in both SS distributions (Figure 5 and Figure 6) and the SVM classification (with three percent difference in accuracies, for instance) (Table 2) were apparent. This implies that there may be few cases, especially in most early AD stage, where each side was diagnosed (by SVM) differently. Unless other historical or clinical data suggest otherwise, it may then be necessary to refer such cases to MRI studies.

Three radiologist readers were given simple instruction to place as few control points and adjust them as required to define hippocampal boundaries. The process was slightly more complicate than measuring, for instance, formation height but less tedious than meticulously tracing them by hand. Since the hippocampal contour were manually defined, and thus different readers may disagree on boundary definitions, as illustrated by two examples (Figure 3). Typically, one would average these shapes per subject to produce a unique consensus. However, it is worth pointing out here that geometrical average on the image space, i.e., Cartesian coordinates, does not necessarily preserve their anatomical realizations. At a given coordinate on the curve, for instance, one reader may identify as hippocampal formation while the other labelled it as choroid fissure. Geometrical mean of these different structures could inevitably result in illegal curve intersection. Prior to obtaining the consensus, the SS was therefore firstly created. To obtain a valid shape consensus, this study then projected the same shape as seen by different readers onto the computed Karhunen-Loeve bases and instead performed averaging on this model space. The averaged model parameters were then back-projected onto the Cartesian coordinates, producing anatomically plausible shape hypothesis for subsequent analyses. In order to further reduce intra-observer variabilities, each reader was also asked to repeat their tracing a few times per subject, the resultant curves were similarly averaged. Note however that, since only two modes variables were considered, including additional modes was anticipated to yield better degree of separation in higher dimensional space. Since both sides were analyzed by separate PCAs and hence spanned with different set of bases, their variation directions may differ. Nonetheless, it can be noticed that the principal directions could capture the change of formation height and choroid fissure width on both sides. This statistical morphology inference concurs with the conventional neurological conjecture of AD manifestation.

On a single slice VCT, AD was generally diagnosed based on hippocampal formation height or area. This approach could have however caused subjective biases due to overall head size and variability of reference axes. To satisfy this argument and thus highlight necessity of involving the proposed statistical shape and SVM analyses, formation areas (averaged per subject) were calculated and their distributions were illustrated by histograms in Figure 7. It is evident that, out of 63 samples, there were 10 (left) and 6 (right) ambiguities found within the overlapping areas. This suggests that only the formation area alone was insufficient to accurately differentiate normal from mild AD.

Among many machine-learning schemes, SVM has been widely applied in medical imaging for several purposes, ranging from segmentation to Computer Aided Diagnoses (CAD). The main advantage of this supervised learning algorithm are that it does not need to hold the Gaussian distribution assumption in the input data and it performs better when only a small number of training samples are available. There are two types of SVMs, i.e., linear and non-linear ones, whose separation of data points is decided by respective type of boundaries. The former is suitable for those datasets that can be easily separated by a hyper-plane whereas the latter is normally used to classify more complex datasets. The concept behind this SVM classifier is to transform this dataset into higher dimensional spaces where they can now be separated using a predefined kernel.

The findings reported in Table 4 imply optimal number of model parameters being dependent on the diagnostic purposes. Specifically, for preliminary screening and confirmation, where a chance of an AD patient being wrongly diagnosed as normal or vice versa should be minimal, the sensitivity and specificity take priority and as such the optimal number of modes included would be seven (the maximum of both sides). Likewise, if precision and accuracy are more important, the number of modes should be that when these values started to converge.

The F-Measure, defined as a harmonic mean between sensitivity and specificity, and could thus be a general compromise. In our experiments on 63 participants, since these convergences occurred almost at the same level of model complexity, it is thus safe to assume that for an early AD diagnosing purpose, the all-round optimal number is seven. Increasing the model complexity to 95% coverage did not improve its performance. On the contrary, as it reached 10 parameters the overfitting started to surface, in which case the SVM tried to perform clustering based on small perturbations at higher dimensions, mostly due to noise. This effect was manifested in the dropping of all statistical measures, as the 10th mode was included. It is, therefore, worth stressing here that, the optimal number of model parameters are not only dependent on faithful model realization, but also on the sample size and on clustering purpose. As a general guideline, one may determine a unified value by averaging these performance indices, and empirically choosing the one that resulted in overall high indices, as illustrated in Figure 8.

## 5. Conclusions

Preliminary screening of normal from early-stage AD by using VCT imaging systems that are available in developing countries is extremely important, since the number of those with Alzheimer's disease is likely to increase over the coming decades. Reliable computational schemes that can assist local physicians to decide whether to refer a case to further detailed MRI study is therefore warranted. It has been shown that early diagnosis of AD can be made by artificial machine learning based on morphological characteristic of hippocampi. Unlike existing structural imaging schemes, the proposed SSA did not require laborious full 3D surface delineation nor any prior knowledge of empirical measures to distinguish AD from normal shapes. In the proposed scheme, SSA served as a means of extracting only dominant variational parameters found in the sampled space, while SVM was adopted to determine discrimination criteria on these parameters to separate two groups. Its main advantage is that a less involving manual task was required, enabling the proposed scheme to be effectively applicable in actual clinical settings for early AD screening of a healthy aging population.

## Figures and Tables

**Figure 1 brainsci-07-00155-f001:**
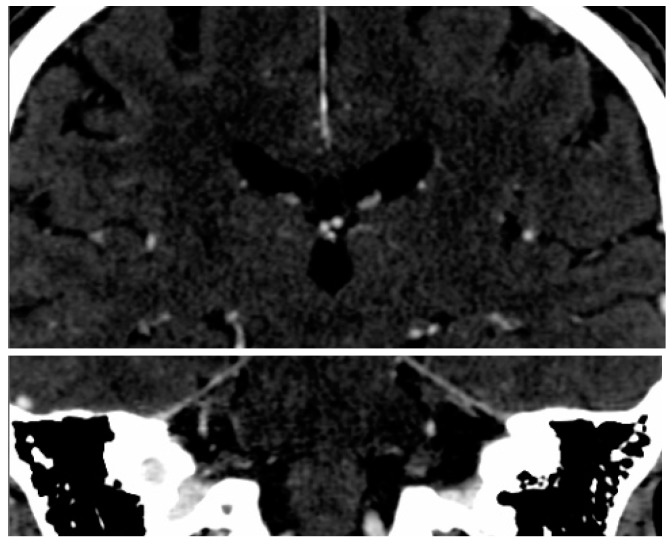
ROI in bilateral hippocampal regions (the white box).

**Figure 2 brainsci-07-00155-f002:**
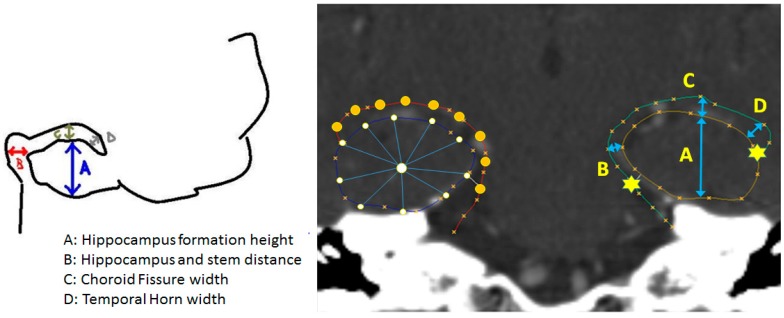
Annotated sketch (**left**) (A = Hippocampal formation height, B = Hippocampal and stem distance, C = Choroid fissure width, D = Temporal horn width) of topological prescription and the corresponding shape overlaid on an image (**right**). For clarity, the fiducial markers (stars) are placed on the left contour, while the regularized control points (circles) are shown on the right one.

**Figure 3 brainsci-07-00155-f003:**
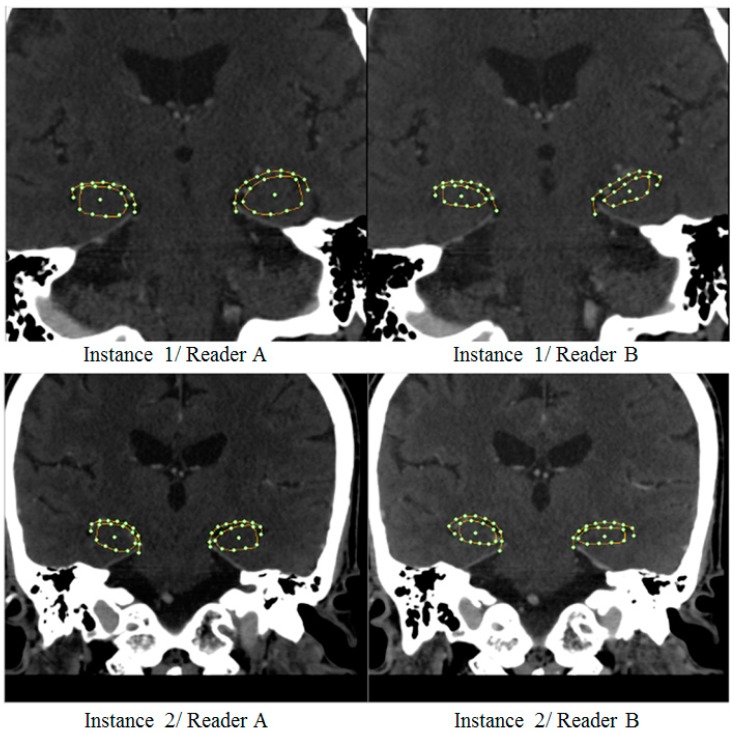
Example of large (**top**) and small (**bottom**) degree of inter-reader variabilities.

**Figure 4 brainsci-07-00155-f004:**
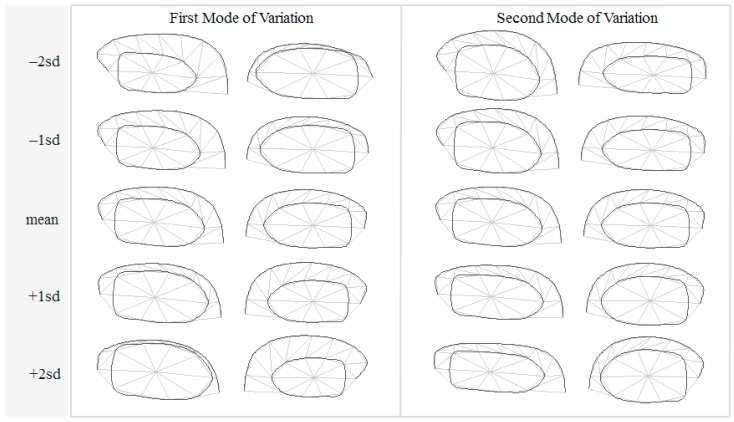
Principal modes of variations within ±2 standard deviations.

**Figure 5 brainsci-07-00155-f005:**
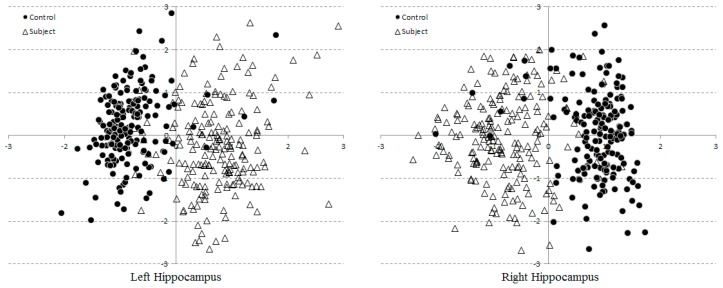
Scatter plots of two model parameters extracted from 382 hippocampi. Horizontal and vertical axes represents the first and second modes of variations, respectively.

**Figure 6 brainsci-07-00155-f006:**
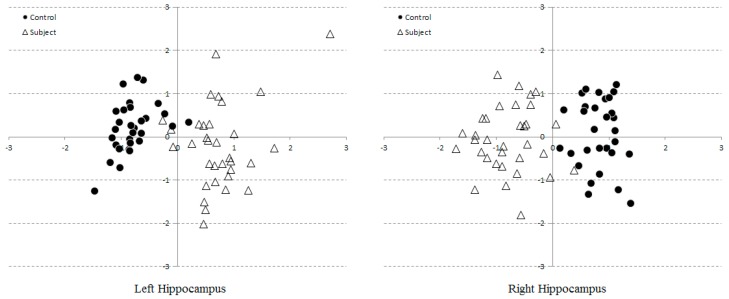
Scatter plots of two model parameters consensually extracted from 63 hippocampi.

**Figure 7 brainsci-07-00155-f007:**
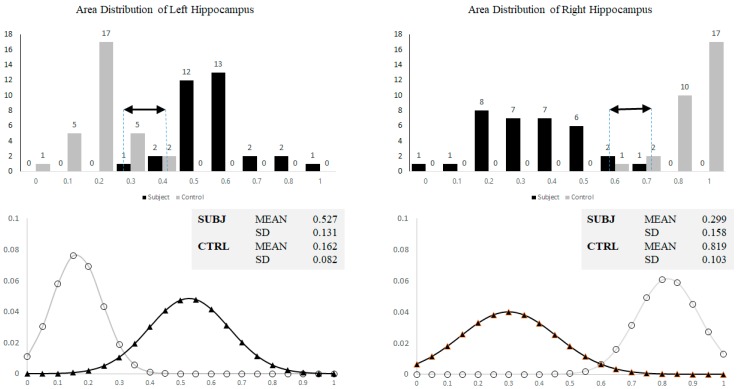
Histograms showing normalized area distribution (**top**) and corresponding approximated Gaussian curves (**bottom**) of left and right hippocampi for both control and subjects. Black arrows indicate overlapping area.

**Figure 8 brainsci-07-00155-f008:**
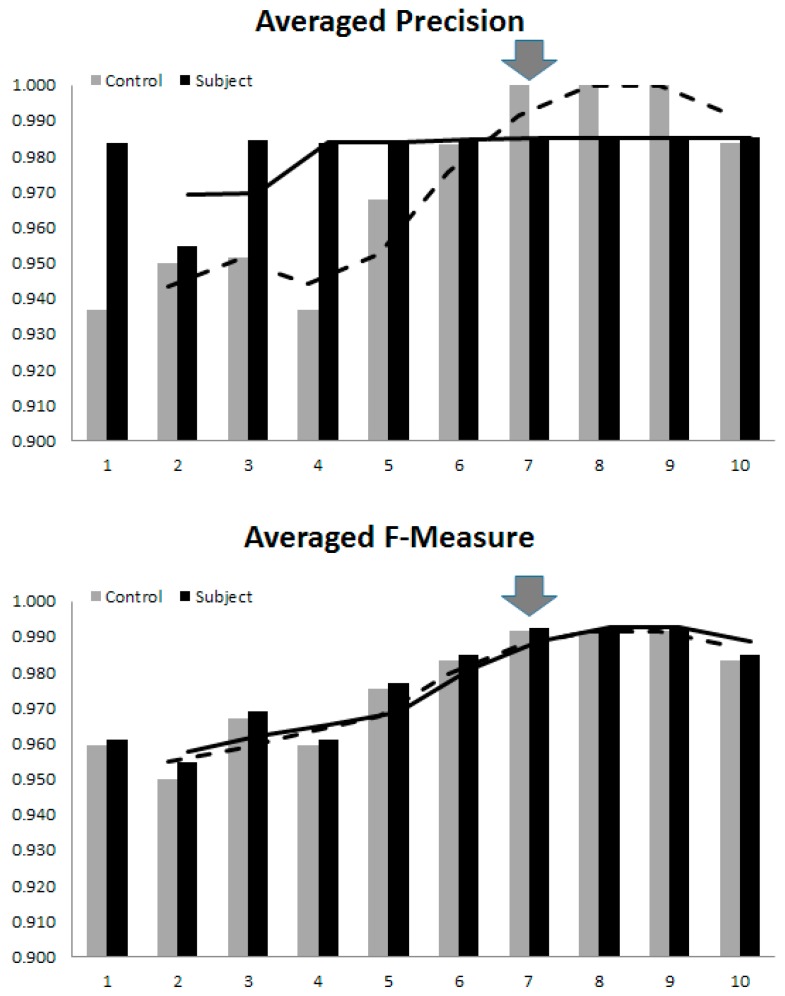
The overall trend of clustering performance versus the model complexity. The arrows indicate the optimal number of model parameters for diagnosing early AD.

**Table 1 brainsci-07-00155-t001:** Characteristic Data.

	Subject *N* = 33	Control *N* = 30	*p*
Age (40–90 Year)	68.51 ± 5.5	67.93 ± 5	0.076
Female	25 (75.8%)	15 (50%)	0.065
Highest level of education	Less than Level 6 (90.9%)	Less than Level 6 (60%)	0.034
Occupation	Retired (75.8%)	Retired (50%)	0.066
Family history of dementia	None	None	-
Average blood pressure (mmHg)	135.1/75.5 ± 14.2/7.3	139.2/81.4 ± 14.4/8	0.064
TMSE* score (point)	18.3 ± 1.6	27.5 ± 1.6	0.027

*p*-Values are assessed using Student *t*-test or Fisher exact tests. * TMSE: Mini-mental State Examination in Thai language.

**Table 2 brainsci-07-00155-t002:** Support Vector Machine (SVM) Numerical Assessments.

Attributes	Left	%	Right	%
Correctly Classified Instances	60	95.2381	62	98.4127
Incorrectly Classified Instance	3	4.7619	1	1.5873
Kappa Statistics	0.9047			0.9682
Mean Absolute Error	0.0476			0.0159
Root Mean Squared Error	0.2182			0.1260
Relative Absolute Error	9.5395%			3.1798%
Root Relative Squared Error	43.6690%			25.2123%

**Table 3 brainsci-07-00155-t003:** Numbers of correctly (TP–C, TP–S, TN–C and TN–S) and incorrectly (FP–C, FP–S, FN–C and FN–S) classified hippocampal samples w.r.t. number of model parameters (modes) taken into account.

**Left**	**Correctly Classified (Samples)**				**Incorrectly Classified (Samples)**			
**Modes**	**TP–C**	**TP–S**	**TN–C**	**TN–S**	**FP–C**	**FP–S**	**FN–C**	**FN–S**
1	29	30	29	30	3	1	1	3
2	28	31	28	31	2	2	2	2
3	29	31	29	31	2	1	1	2
4	29	30	29	30	3	1	1	3
5	29	31	29	31	2	1	1	2
6	29	32	29	32	1	1	1	1
7	29	33	29	33	0	1	1	0
8	29	33	29	33	0	1	1	0
9	29	33	29	33	0	1	1	0
10	29	33	29	33	0	1	1	0
**Right**	**Correctly Classified (Samples)**				**Incorrectly Classified (Samples)**			
**Modes**	**TP-C**	**TP-S**	**TN-C**	**TN-S**	**FP-C**	**FP-S**	**FN-C**	**FN-S**
1	30	32	30	32	1	0	0	1
2	29	32	29	32	1	1	1	1
3	30	32	30	32	1	0	0	1
4	30	32	30	32	1	0	0	1
5	30	33	30	33	0	0	0	0
6	30	33	30	33	0	0	0	0
7	30	33	30	33	0	0	0	0
8	30	33	30	33	0	0	0	0
9	30	33	30	33	0	0	0	0
10	30	32	30	32	1	0	0	1

It can be noticed that the number of incorrectly classified samples decreased, as more number of modes were considered. For the left side, accuracy was not improved after 7 modes, while for the right one, the classification was consistently accurate for the 5–9 modes, after which a sign of over fitting started to appear. In addition, to evaluate the model performance, various aspects of the SVM classification, i.e., sensitivity, specificity, precision, accuracy, and F-measure were computed for each side of hippocampus (Table 4).

**Table 4 brainsci-07-00155-t004:** Sensitivity, specificity, precision, accuracy and F-measure for the controls (C) and subjects (S) groups. The results w.r.t. the number of modes (M) of both left (L) and right (R) sides are shown.

**L**	**Sensitivity**		**Specificity**		**Precision**		**Accuracy**		**F-Measure**	
**M**	**C**	**S**	**C**	**S**	**C**	**S**	**C**	**S**	**C**	**S**
1	0.967	0.909	0.906	0.968	0.906	0.968	0.935	0.938	0.935	0.938
2	0.933	0.939	0.933	0.939	0.933	0.939	0.933	0.939	0.933	0.939
3	0.967	0.939	0.935	0.969	0.935	0.969	0.951	0.954	0.951	0.954
4	0.967	0.909	0.906	0.968	0.906	0.968	0.935	0.938	0.935	0.938
5	0.967	0.939	0.935	0.969	0.935	0.969	0.951	0.954	0.951	0.954
6	0.967	0.970	0.967	0.970	0.967	0.970	0.967	0.970	0.967	0.970
7	0.967	1	1	0.971	1	0.971	0.983	0.985	0.983	0.985
8	0.967	1	1	0.971	1	0.971	0.983	0.985	0.983	0.985
9	0.967	1	1	0.971	1	0.971	0.983	0.985	0.983	0.985
10	0.967	1	1	0.971	1	0.971	0.983	0.985	0.983	0.985
**R**	**Sensitivity**		**Specificity**		**Precision**		**Accuracy**		**F-Measure**	
**M**	**C**	**S**	**C**	**S**	**C**	**S**	**C**	**S**	**C**	**S**
1	1	0.970	0.968	1	0.968	1	0.984	0.985	0.984	0.985
2	0.967	0.970	0.967	0.970	0.967	0.970	0.967	0.970	0.967	0.970
3	1	0.970	0.968	1	0.968	1	0.984	0.985	0.984	0.985
4	1	0.970	0.968	1	0.968	1	0.984	0.985	0.984	0.985
5	1	1	1	1	1	1	1	1	1	1
6	1	1	1	1	1	1	1	1	1	1
7	1	1	1	1	1	1	1	1	1	1
8	1	1	1	1	1	1	1	1	1	1
9	1	1	1	1	1	1	1	1	1	1
10	1	0.970	0.968	1	0.968	1	0.984	0.985	0.984	0.985

From Table 4, the model sensitivity increased as the number of modes increased. For the left hippocampus, the sensitivity converged at the 3rd and 7th mode for control and subject groups, respectively. Likewise, for the right hippocampus the value converged at the 3rd and 5th modes. The model specificity, precision, accuracy and F-measure of the left hand side also increased with the number of models and converged at the 7th mode for both controls and subjects. For the right hand side, the similar trend can be observed with convergence occurred starting from the 3rd and not later than the 5th mode for both groups.

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
