# Peer review of "Hyperplanar Morphological Clustering of a Hippocampus by Using Volumetric Computerized Tomography in Early Alzheimer’s Disease"

_brainsci, 2017, doi:10.3390/brainsci7110155_

Round 1

Reviewer 1 Report

Bi-laterality and hemispheric asymmetry widely recognized as the in-dispensable evolutionally conserved attributes of the brain (for review see (Dyakin at al. 2017). In particular the hemispheric asymmetry of the healthy human hippocampus is well established (Gonçalves-Pereira et al., 2006; Woolard & Heckers. 2012).

Giving the significance of above mentioned;

The above mentioned references (among others) must be included in the text of paper.

The absence of hippocapmal asymmetry in the current experiment should be explained. It could be the limited sensitivity of protocol or the specificity of  participant group (sex, aging, and others).

REFERENCES

(Gonçalves-Pereira et al., 2006) P.M. Gonçalves-Pereira, E. Oliveira, R. Insausti. Quantitative volumetric analysis of the hippocampus, amygdala and entorhinal cortex: normative database for the adult Portuguese population. Rev Neurol. 2006 Jun 16-30;42(12):713-22.

(Woolard & Heckers. 2012) A. Woolard and S. Heckers. Anatomical and functional correlates of human hippocampal volume asymmetry. Psychiatry Res. (2012) 201(1): 48–53. Doi:  10.1016/j.pscychresns.2011.07.016.

(Dyakin at al. 2017) V. V. Dyakin, J. Lucas, N. V. Dyakina-Fagnano, E. V. Posner and C. Vadasz. Chain of Chirality Transfer as Determinant of Brain Functional Laterality. Breaking the Chirality Silence: Search for New Generation of Biomarkers. Relevance to Neurodegenerative Diseases, Cognitive Psychology and Nutrition Science. NNR (2017). Doi:10.24983/scitemed.nnr.

Author Response

Dear Editor,

Cc: the Reviewers

In keeping with our previous correspondence regarding the revision due on November 6, we are resubmitting our revised version of the manuscript brainsci-232372, Hyperplanar Morphological Clustering of a Hippocampus by using Volumetric Computerized Tomography in Early Alzheimer’s Disease, for the Brain Sciences Journal.

We thank the reviewers for constructive critiques on our manuscript. Valuable time and details provided by each reviewer are greatly appreciated. Their feedback has been thoroughly considered and addressed, and suggested changes have been incorporated into the revised manuscript.

Please kindly find in following pages, the point-by-point reviewers' specific comments and our response/ revisions.

Sincerely Yours,

Sarawut Suksuphew

Comments

Responses

Bi-laterality   and hemispheric asymmetry widely recognized as the in-dispensable   evolutionally conserved attributes of the brain (for review see (Dyakin at   al. 2017). In particular the hemispheric asymmetry of the healthy human   hippocampus is well established (Gonçalves-Pereira et al., 2006; Woolard   & Heckers. 2012).

Giving   the significance of above mentioned;

The   significance of this statement are included (L282-287, P11)

The   above mentioned references (among others) must be included in the text of   paper.

The   references have been included (L447-455, P15).

The   absence of hippocampal asymmetry in the current experiment should be   explained. It could be the limited sensitivity of protocol or the specificity   of participant group (sex, aging, and others).

The   absence of notable asymmetry in this study is explained (L288-293, P11)

Reviewer 2 Report

Work performed on imaging systems which are available in developing countries is extremely important. This is particularly important in Alzheimer's disease since the number of those with this disease is likely to increase over the coming decades. This makes work such as that described in this paper extremely timely.

However, the use of English is very poor and it is advised that the authors work with an English speaker (particularly one who has published scientific papers previously) to redraft the paper. This should make understanding of the whole reason for doing the work (introduction), how the work was done (methods), results and discussion much easier to understand. In its current form, the paper is very hard to read.

Further to this, are the authors not complicating matters by asking individuals to outline the hippocampus? Why not visually rate the hippocampal atrophy (as per Scheltens' scale). Does this provide as good a discrimination between AD and controls? Alternatively, why not just provide a volume? Is the complex SVM better than a volume?

How did the authors deal with the fact that the groups are not equivalent (such as for education). It is recommended that the authors seek help from a statistician to discuss whether adjustment for such variables is necessary. A statistician could also advise as to the best way to show data (such as fig 5) - shouldn't there be a single point per subject?

Author Response

Dear Editor,

Cc: the Reviewers

In keeping with our previous correspondence regarding the revision due on November 6, we are resubmitting our revised version of the manuscript brainsci-232372, Hyperplanar Morphological Clustering of a Hippocampus by using Volumetric Computerized Tomography in Early Alzheimer’s Disease, for the Brain Sciences Journal.

We thank the reviewers for constructive critiques on our manuscript. Valuable time and details provided by each reviewer are greatly appreciated. Their feedback has been thoroughly considered and addressed, and suggested changes have been incorporated into the revised manuscript.

Please kindly find in following pages, the point-by-point reviewers' specific comments and our response/ revisions.

Sincerely Yours,

Sarawut Suksuphew

Comments

Responses

Work   performed on imaging systems which are available in developing countries is   extremely important. This is particularly important in Alzheimer's disease   since the number of those with this disease is likely to increase over the   coming decades. This makes work such as that described in this paper   extremely timely.

This   observation has been included. (L359-362, P13)

However,   the use of English is very poor and it is advised that the authors work with   an English speaker (particularly one who has published scientific papers   previously) to redraft the paper. This should make understanding of the whole   reason for doing the work (introduction), how the work was done (methods),   results and discussion much easier to understand. In its current form, the   paper is very hard to read.

The   language editing has been made (highlighted in green).

Further   to this, are the authors not complicating matters by asking individuals to   outline the hippocampus? Why not visually rate the hippocampal atrophy (as   per Scheltens' scale). Does this provide as good a discrimination between AD   and controls? Alternatively, why not just provide a volume? Is the complex   SVM better than a volume?

Three   radiologist readers were given simple instruction to place as few control   points and adjust them as required to define hippocampal boundaries. The   process was slightly more complicate than measuring, for instance, formation   height but less tedious than meticulously tracing them by hand. (L294-296,   P11).

The   reason for not choosing analog Schelten’s scale are provided (L45-50, P1-2).

Since   only a single slice was considered, AD volume could not be evaluated.   However, to highlight the merits of the proposed SSA and SVM scheme, it was   benchmarked against conventional diagnosis based on formation area (L175-182,   P5), whose results are provided (Figure 7) and pros and cons are   discussed (L316-322, P11).        

How   did the authors deal with the fact that the groups are not equivalent (such   as for education). It is recommended that the authors seek help from a   statistician to discuss whether adjustment for such variables is necessary. A   statistician could also advise as to the best way to show data (such as fig   5) - shouldn't there be a single point per subject?

The   subject selection protocol has taken into account different educational   background (L108, P3). The statistical of subject characteristics are   provided (L197-198, P5).

In   Figure 5, each point corresponds to a shape instance drawn by one reader on   one subject. A reader might repeat his/ her delineations for any subject a   couple times, depending on his/ her confidence (L221-223, P7). This   resulted in 382 points in total.

These   instances were averaged the readings per subject in the model space (L231,   P7 and L304-309, P11), resulting in a single point per subject as   shown in Figure 6.
